# Investigating the Antidepressant Mechanisms of *Polygonum sibiricum* Polysaccharides via Microglial Polarization

**DOI:** 10.3390/nu16030438

**Published:** 2024-02-01

**Authors:** Yingyu Zhang, Danyang Wang, Jiameng Liu, Jing Sun, Xinmin Liu, Bei Fan, Cong Lu, Fengzhong Wang

**Affiliations:** 1Institute of Food Science and Technology, Chinese Academy of Agricultural Sciences (CAAS), Beijing 100193, Chinaljiam88@163.com (J.L.);; 2College of Food Science and Engineering, Shanxi Agricultural University, Jinzhong 030801, China; 3Institute of New Drug Technology, Ningbo University, Ningbo 315211, China

**Keywords:** *Polygonum sibiricum* polysaccharides, neuroinflammation, microglia polarization

## Abstract

*Polygonum sibiricum*, with its medicinal and edibility dual properties, has been widely recognized and utilized throughout Chinese history. As a kind of its effective component, *Polygonum sibiricum* polysaccharides (PSP) have been reported to be a promising novel antidepressant agent. Meanwhile, the precise mechanisms underlying its action remain elusive. The polarization state transition of microglia is intricately linked to neuroinflammation, indicating its crucial involvement in the pathophysiology of depression. Researchers are vigorously pursuing the exploration of this potential treatment strategy, aiming to comprehend its underlying mechanisms. Hence, the current study was designed to investigate the antidepressant mechanisms of PSP via Microglial M1/M2 Polarization, based on the lipopolysaccharide (LPS)-induced BV2 cell activation model. The results indicate that PSP significantly inhibited NO and LDH release and reduced ROS levels in LPS-induced BV2 cells. PSP could significantly reduce the protein expression level of Iba-1, decreased the mRNA levels of TNF-α, IL-1β, and IL-6, and increased the mRNA level of IL-10. PSP also significantly reduced the protein expression level of CD16/32 and increased that of CD206, reduced the mRNA level and fluorescence intensity of iNOS, and increased those of Arg-1. However, PSP pretreatment reversed the alterations of the BDNF/TrkB/CREB and Notch/Hes1 pathways in LPS-induced BV2 cells. These results suggested that PSP exerted the anti-inflammatory effects by inhibiting M1 phenotype polarization and promoting microglia polarization toward the M2 phenotype, and its regulation of microglia M1/M2 polarization may be associated with modulating the BDNF/TrkB/CREB and Notch/Hes1 pathways.

## 1. Introduction

In recent years, depression has emerged as a significant global health concern, with increasing prevalence and impact on individuals and communities worldwide. Depression will rise from the third highest global burden of diseases to the first by 2030 [1]. The late COVID-19 pandemic has caused the prevalence to rise globally, especially in middle- and low-income countries [2]. The biological mechanisms of depression are complex and mainly related to the multiple different systems and targets [3], which pose great difficulties to the development of drugs used to treat depression. Frontline antidepressants not only require a considerable amount of time to produce therapeutic effects, but their induced side effects such as sexual dysfunction, weight gain, nausea, and headaches are often undeniable [4,5]. Therefore, it is urgent to discover new methods and targets for the treatment of depression.

Patients with depression usually experience an increase in systemic inflammation and neuroinflammation, and the inflammatory process is related to the pathophysiology of depression [6,7]. The inflammatory response is the result of immune system activation. Increasing evidence suggests that there are different types of immune cell behavior imbalances in mental illnesses, including depression. Immune cells release cytokines, chemokines, and secondary inflammatory mediators such as prostaglandins in response to inflammatory stimuli [8]. Microglia play a strong part in various physiological processes of the central nervous system, such as immunity, inflammation, synaptic transmission, neural plasticity, the maintenance of neural networks, and tissue damage repairments [9]. The behavioral imbalance of microglia is considered as an important trigger for the inflammatory response in depression [10]. However, studies have shown that by regulating the polarization phenotype of microglia, which is usually simplified as the M1/M2 polarization phenotype, it is possible to antagonize the behavioral imbalance of microglia, counteract inflammatory damage, and promote the recovery and remodeling of neural tissue [11]. Therefore, the modulation of overactivated inflammatory M1 microglia to the protective M2 phenotype is considered as one of the potential therapeutic targets of depression [12].

*Polygonum sibiricum* (PS) is attached to the Liliaceae family. Its medicinal and edibility dual properties have been widely recognized and utilized in various applications throughout Chinese history. PS has a highly diverse composition of bioactive compounds such as flavones, homoisoflavanone, alkaloids, lignins, steroid saponins, triterpenoid saponins, polysaccharides, etc., and health-improving activities such as anti-aging, anti-inflammatory, anti-osteoporosis, enhancing immunity, improving sleep, etc. [13]. Among them, *Polygonum sibiricum* polysaccharides (PSP) are a kind of its effective component, showing the antidepressant activity. The antidepressant effects of PSP have been confirmed in lipopolysaccharides (LPS)-induced mice, which not only reversed depressive symptoms, but also mitigated oxidative stress, inflammation, and hypothalamic–pituitary–adrenal (HPA) axis hyperfunction in mice [14]. The antidepressant effects of PSP were also related to regulate the oxidative stress–calpain-1-NLRP3 signaling axis [15]. PSP also holds the potential to enhance the performance of mice experiencing acute despair in rigorous behavioral tests, which may be related to reducing the monoamine neurotransmitter levels, inhibiting the release of inflammatory cytokines, and regulating tryptophan metabolism [16]. PSP could restore beneficial gut microbiota, improve intestinal barrier integrity, and exert antidepressant effects through the microbiota–gut–brain axis [17,18]. Meanwhile, it is unclear whether PSP could inhibit the activation of microglia and promote M2 phenotype polarization, and its related molecular mechanisms still need further research.

Therefore, to gain a deeper understanding of the antidepressant properties of PSP, this study delved into its potential impact on microglial polarization in the LPS-induced BV2 cell activation model. Additionally, the study explored the signaling pathways that are involved in both depression and microglial polarization. The goal is to offer a fresh perspective on the clinical potential of PSP as an antidepressant by focusing on the role of microglial polarization and the regulation of correlated signaling pathways.

## 2. Materials and Methods

### 2.1. Materials

PSP contained 72.12% carbohydrate (Mw = 20.48 KDa) which was obtained from Shanghai Yuanye Bio-Technology Co., Ltd. (Shanghai, China). Infrared spectroscopy showed that the PSP samples had typical polysaccharide structure characteristics.

The BV2 mouse microglial line was purchased from Wuhan Sunncell Bio-Technology Co., Ltd. (Wuhan, China); fetal bovine serum (FBS) and Dulbecco’s modified Eagle’s medium (DMEM) from Gibco (Grand Island, NE, USA); lipopolysaccharides (LPS) from Sigma (St. Louis, MO, USA); enzyme-linked immunosorbent assay (ELISA) kits for tumor necrosis factor (TNF-α), interleukin (IL)-6, IL-1β, and IL-10 from Jianglai Biological Technology Co., Ltd. (Shanghai, China); and the Nitric Oxide (NO) assay kit, CCK-8 assay kit, and DCFH-DA Reactive Oxygen (ROS) Fluorescent Probe from Solarbio Science and Technology Co., Ltd. (Beijing, China). Antibodies targeting iNOS, BDNF, ERK, Phospho-ERK, CREB, Phospho-CREB, TrkB, and Phospho-TrkB from Abcam (Cambridge, UK); antibody targeting Arginase-1 from Cell Signaling Technology (Boston, MA, USA); antibodies targeting Iba-1, Notch1, and Hes1 from Santacruz Biotechnology (Santa Cruz, CA, USA); goat anti-rabbit AlexFluor 488^®^ secondary antibodies from Servicebio Co., Ltd. (Shanghai, China); and CD16/CD32 monoclonal antibody PE and CD206 (MMR) monoclonal antibody PE from Ebioscience (San Diego, CA, USA).

### 2.2. Cell Culture and Treatment

The BV2 mouse microglial cell lines were cultured in DMEM medium containing 10% FBS and 5% CO_2_, at 37 °C. Cells were passaged every 24 h. After 24 h, the experiment was performed with the control group, LPS group (1 μg/mL), and PSP groups (400, 800, and 1000 μg/mL). PSP was added to preactivate for 24 h, and then 1 μg/mL of LPS was added to co-stimulate for 12 h.

### 2.3. Cell Viability Assay

BV2 cells were seeded in a 96-well plate at a density of 5 × 10^3^ cells/well. After 12 h, cells were pretreated with different concentrations of PSP for 24 h, followed by incubation with or without LPS for 12 h. Additionally, 10 μL of CCK-8 solution was added to each well. After incubation for 1 h at 37 °C, the absorbance at 450 nm was measured using a microplate reader.

### 2.4. NO Assay

In the logarithmic growth phase, BV2 cells were seeded into a 24-well culture plate at a density of 2.5 × 10^4^ cells per well. After 12 h, cells were pretreated with different concentrations of PSP for 24 h. Then, cells were incubated with or without LPS for 24 h. The NO contents in the supernatant were determined following NO assay kit protocol (Beyotime Biotechnology, Shanghai, China).

### 2.5. ROS Assay

BV2 cells were cultured overnight in a 24-well culture plate at a density of 5 × 10^4^ cells/well, then exposed to different concentrations of PSP for 24 h. Subsequently, cells were incubated with or without LPS for 6 h. The cells were stained with a 4 μM DCFH-DA fluorescent probe (Solarbio Science and Technology, China) for 10 min at 37 °C and washed with PBS. All images were observed with fluorescence microscopy (Olympus, Japan) and analyzed by Image J v1.5.4 software (National Institutes of Health, Bethesda, MD, USA).

### 2.6. Quantitative PCR (qPCR) Assay

BV2 cells were seeded into a 6-well culture plate at a density of 1 × 10^5^ cells/well. After 12 h, cells were pretreated with different concentrations of PSP for 24 h, and incubated with or without LPS for 12 h. RNA was isolated according to the EASYspinPlus rapid tissue/cellular RNA extraction kit (Aidlab Biotechnologies, Beijing, China). The GoScript Reverse Transcription System (Promega Biotech, Beijing, China) was used for the synthesis of cDNA. qRT-PCR was performed in the Applied Biosystems QuantStudio 7 Flex system (Thermo Fisher Scientific, Waltham, MA, USA) using the NovoStart SYBR qPCR SuperMix Plus kit (Novoprotein, Beijing, China). The amplification parameters were 95 °C for 1 min, followed by 40 cycles of 95 °C for 2 s and 60 °C for 1 min. The relative gene expression was calculated using the 2^−ΔΔCT^ method and normalized with the control gene β-actin. The primers’ information is shown in Table 1.

### 2.7. Enzyme-Linked Immunosorbent Assay (ELISA)

BV2 cells were seeded into a 6-well culture plate at a density of 1 × 10^5^ cells/well. After 12 h, cells were pretreated with different concentrations of PSP for 24 h, and incubated with or without LPS for 12 h. The supernatant was collected to measure the levels of TNF-α, IL-1β, IL-6, and IL-10 using commercial ELISA kits (Jianglai Biological Technology, Shanghai, China) based on the manufacturer’s instructions.

### 2.8. Immunofluorescence Staining

BV2 cells were seeded in a 6-well culture plate at a density of 1 × 10^5^ cells/well, and cultured and treated as before mentioned. BV2 cells were fixed with 4% paraformaldehyde for 10 min at room temperature, then permeabilized using 0.5% Triton X-100 for 30 min on ice. The cells were blocked with 5% BSA for 1 h. The cells were incubated with iNOS (1:50, ab283655) and Arg-1 (1:50, #93668) primary antibodies at 4 °C overnight. On the following day, cells were washed with PBS and incubated with AlexFluor 488^®^ (1:500) secondary antibodies at 37 °C for 1 h. To preserve the fluorescent signal, the cells were treated with an anti-fade mounting medium with DAPI for 10 min at 37 °C. All images were observed with fluorescence microscopy (Olympus, Tokyo, Japan) and analyzed by Image J software (National Institutes of Health, USA).

### 2.9. Flow Cytometry

BV2 cells in a 6-well culture plate were cultured and treated as indicated. Then, the cells were washed and re-suspended in cold PBS at a density of 1 × 10^6^ cells/mL. Subsequently, 0.125 μg of anti-CD16/CD32 PE (93) and anti-CD206 PE (MR6F3) was added separately, and then incubated at room temperature in the dark for 30 min. The cells were washed twice with PBS and re-suspended in 400 μL of PBS. The samples were analyzed using flow cytometry (CytoFLEX, Indianapolis, IN, USA).

### 2.10. Western Blot Analysis

BV2 cells were seeded in a 6-well culture plate at a density of 1 × 10^5^ cells/well, and cultured and treated as mentioned above. BV-2 microglia were washed with precooled PBS. The samples were collected by centrifugation (1000 rpm, 5 min). Then, 5 times the volume of RIPA lysis buffer (containing PMSF) was added to the cell precipitate and incubated on ice for 10 min, followed by centrifugation (12,000× *g*, 10 min, 4 °C) to collect the supernatant. The protein concentration was determined by the BCA Kit (Beyotime, Nanjing, China). According to the protein quantitative results, the protein concentrations were unified. Proteins were separated by means of sodium dodecyl sulphate-polyacrylamide gel electrophoresis (SDS-PAGE) and then transferred to polyvinylidene fluoride (PVDF) membranes for a duration of 120 min at a constant current of 300 mA. The membranes were sequentially incubated with primary (BDNF, ab108319; ERK, ab32537; CREB, ab32515; Phospho-CREB, ab32096; TrkB, ab187041; Phospho-TrkB, ab229908; Notch1, sc-376403-1; Hes1, sc-166410-1) and secondary antibodies and enhanced chemiluminescence (ECL) solutions, followed by autoradiography. The intensity of the blot was analyzed using Image pro plus 6.0.

### 2.11. Statistical Analysis

Statistical analysis was performed using SPSS 23.0, GraphPad Prism 8.0, and ImageJ software, and the results were expressed as the mean ± standard error (Mean ± SEM). Multiple group comparisons were performed using one-way ANOVA, while pairwise comparisons between groups were performed using the Least Significant Difference (LSD) method. *p* < 0.05 was considered statistically significant.

## 3. Results

### 3.1. Effects of PSP on the Viability of BV2 Cells and Microglial Activation

The effects of various concentrations of PSP on BV2 cell viability were shown in Figure 1A. Notably, PSP concentrations ranging from 100 to 1000 μg/mL did not diminish the viability of BV2 cells after 24 h treatment. PSP did not induce any detectable cytotoxicity compared to the control group. The combined effects of PSP and LPS on BV2 cell viability were shown in Figure 1B. The CCK-8 results indicated that PSP at different concentrations (400, 800, 1000 μg/mL) with or without LPS (1 μg/mL) had no significant toxic effects on cell viability. Thus, 400, 800, and 1000 μg/mL of PSP were chosen for subsequent experiments.

Iba-1 is a marker protein of microglia. To evaluate the effects of PSP on LPS-induced microglial activation, the protein expression levels of Iba-1 in BV2 cells were evaluated. As shown in Figure 1C,D, LPS induced microglial activation. The protein expression level of Iba-1 was significantly promoted after LPS stimulation (*p* < 0.001). However, PSP intervention significantly reduced the protein expression level of Iba-1 (*p* < 0.05, *p* < 0.001). These results indicated that PSP pretreatment could reverse LPS-induced microglial activation.

### 3.2. Effects of PSP on Intracellular Reactive Oxygen Species in LPS-Induced BV2 Cells

The fluorescence intensity of the LPS group was substantially higher than the control group (Figure 2, *p* < 0.001). LPS stimulation increased ROS production in BV2 cells. Compared to the LPS group, PSP (200, 400, 800, 1000 μg/mL) pretreatment significantly reduced fluorescence intensity (*p* < 0.01, *p* < 0.001), indicating that PSP could inhibit LPS-induced oxidative damage in BV2 cells to a certain extent.

### 3.3. Effects of PSP on the Production of NO and Inflammatory Cytokine in LPS-Induced BV2 Cells

It is shown that the NO level was significantly elevated after LPS treatment compared to the control group (Figure 3A). PSP (400, 800, 1000 μg/mL) reduced NO levels in a dose-dependent manner. Both qRT-PCR and ELISA were used to determine the effects of PSP on the expression and release of inflammatory factors in LPS-induced BV2 cells. The qRT-PCR results (Figure 3B–G) showed that LPS stimulation significantly increased the expression level of M1 pro-inflammatory cytokines (TNF-α, IL-1β, IL-6, and iNOS) and decreased M2 anti-inflammatory cytokines (Arg-1) (*p* < 0.001). PSP intervention significantly decreased the mRNA levels of M1 markers (TNF-α, IL-1β, IL-6, and iNOS) and increased the M2 marker (IL-10, Arg-1) (*p* < 0.05, *p* < 0.01, *p* < 0.001). In agreement with the results obtained from q-PCR, ELISA (Figure 3H–K) showed that treatment with LPS significantly increased the level of pro-inflammatory cytokines (TNF-α, IL-1β, and IL-6). Conversely, the level of the anti-inflammatory cytokine (IL-10) was notably reduced following LPS exposure (*p* < 0.001). These changes were reversed in the presence of PSP (*p* < 0.001). PSP has the potential to stimulate the polarization of BV2 cells from the M1 to M2 phenotype and inhibit the release of pro-inflammatory cytokines, exerting the anti-inflammatory effects.

### 3.4. PSP Motivated Microglial Polarization to the M2 Phenotype in LPS-Induced BV2 Cells

M1 polarization in microglia was represented by CD16/32 and iNOS, while CD206 and Arg-1 are markers of M2 polarization. The effects of PSP on the polarization phenotype of BV2 cells were analyzed by flow cytometry and immunofluorescence staining. Firstly, the protein expression levels of CD16/32 and CD206 were measured by flow cytometry. As shown in Figure 4, LPS treatment significantly increased the protein expression of CD16/32 and reduced CD206 (*p* < 0.001). The protein expression of CD16/32 was markedly downregulated and the CD206 was upregulated by PSP pretreatment compared to the LPS group (*p* < 0.05, *p* < 0.001). Then, the expression of iNOS and Arg-1 were further measured by immunofluorescence staining. As shown in Figure 5, the fluorescence intensity of iNOS in the LPS group was significantly increased (*p* < 0.001), and PSP intervention significantly reduced the fluorescence intensity of iNOS (*p* < 0.05, *p* < 0.01, *p* < 0.001). Although the fluorescence intensity of Arg-1 in the LPS group did not significantly change compared with the control group, PSP pretreatment obviously enhanced the immunoreactivity of Arg-1 (*p* < 0.05, *p* < 0.001). These results suggested that PSP inhibited microglial polarization to the M1 phenotype and promoted microglial polarization to the M2 phenotype.

### 3.5. Effects of PSP on BDNF/TrkB/CREB Signaling Pathway in LPS-Induced BV2 Cells

It is shown that LPS remarkably decreased the expressions of BDNF, p-TrkB/TrkB, and p-CREB/CREB in BV2 cells when compared to those of the control group (*p* < 0.01, *p* < 0.001) (Figure 6). Meanwhile, PSP pretreatment obviously reversed these changes (*p* < 0.05, *p* < 0.01, *p* < 0.001). The results demonstrated that PSP could regulate the BDNF/TrkB/CREB pathway, which may probably contribute to the regulation of microglial polarization.

### 3.6. Effects of PSP on Notch Signaling Pathway in LPS-Induced BV2 Cells

As shown in Figure 7, we measured the expressions of Notch1 and Hes1. The results indicated that the expressions of Notch1 and Hes1 were remarkably increased in the LPS group when compared with that of the control group (*p* < 0.001). However, PSP pretreatment obviously reversed these changes (*p* < 0.05, *p* < 0.001). It is indicated that PSP regulated the Notch pathway, which was associated with the regulation of microglial polarization.

## 4. Discussion

Recently, a growing body of research has increasingly emphasized the critical role of neuroinflammation in the proper function of the central nervous system (CNS), such as the stress response, emotion and cognitive activities [19], and inflammation, which has a dual effect on the CNS. Acute inflammatory reactions help maintain body balance, eliminate necrotic and damaged cells, initiate tissue repair, and exert protective effects. However, excessive inflammatory reactions often release multiple inflammatory and toxic factors, resulting in toxic damage and various diseases, including depression. Among them, the imbalance of the M1/M2 polarization of microglia is considered to be the key factor affecting the severity of neuroinflammation. The M1 phenotype leads to a chronic neuroinflammatory reaction, induces the release of proinflammatory cytokines such as TNF-α and IL-1β, exacerbates tissue damage, and further aggravates the pathological process of depression. The M2 phenotype, releasing anti-inflammatory factors such as IL-4 and IL-10 and some neurotrophic factors, plays a positive role in damage repair and functional recovery [20]. Therefore, current research believes that in the treatment of neuroinflammatory-related diseases, inhibiting the M1 phenotype while activating M2 phenotype polarization may bring more significant overall effects than inhibiting M1-microglia alone.

The M1/M2 phenotype of microglia represents two extremes of the neuroinflammatory phenotype. Studies show that different phenotypes can also be divided into different subtypes; for example, the M2 phenotype can also be divided into M2a, M2b, M2c, and M2d subtypes [21], although these are not completely independent activation states. Therefore, the simple classification of microglia into the classic activated-M1 phenotype and alternative activated-M2 phenotype is the most common method to clarify the role of microglia in the CNS. At present, the identification of the M1/M2 phenotype mainly focuses on the following three aspects: cell surface specific molecule expression, the cell secretion of inflammatory and anti-inflammatory factors, and arginine pathway metabolites [22,23,24,25]. Based on these reports and combined with the previous relevant research, our study selected iNOS, TNF-α, IL-1β, IL-6, and CD16/32 as M1 phenotype markers and Arg-1, IL-10, and CD206 as M2 phenotype markers.

Most work on microglial activation and signaling has been performed in vitro, frequently by using cell lines such as BV-2 [26]. The immortality mouse BV2 cell line retains many morphological, phenotypic, and functional characteristics of microglia [27]. LPS is the inflammatory response inducer and a classic activator of microglia [28]. It can induce 90% of the gene expressions in BV-2 cells and showed reaction patterns similar to primary microglia, which made the cell line a suitable experimental model [29]. In rodent models, the intraperitoneal injection of LPS is a classic depression model based on the neuroinflammation hypothesis. It can induce depressive-like behaviors such as a lack of pleasure and reduced social and exploratory activities in rodents, and also result in the excessive activation of microglia [30]. At present, the LPS-induced BV2 cell activation model has been the most commonly used model for studying the M1/M2 polarization regulation of active ingredients in vitro [31,32]. Therefore, the BV2 cell line and LPS-induced activation model were selected in the current study. In agreement with the previous research [33], the NO and ROS levels and expression level of the microglial marker protein Iba-1 were significantly increased after being exposed to LPS, indicating that the BV2 cells were activated. The mRNA levels and supernatant contents of the M1 phenotype markers (iNOS, TNF-α, IL-1β, IL-6) and protein expression of CD16/32 were markedly increased. However, the mRNA levels and supernatant contents of the M2 phenotype markers (IL-10, Arg-1) and protein expression of CD206 were significantly decreased. These results demonstrated that LPS induced the transition of BV2 cells to the M1 phenotype, and the neuroinflammation model in vitro was successfully constructed.

Based on that the antidepressant and anti-inflammation effects of PSP have been reported, the current study was mainly focused on the role of PSP in the modulation of microglial polarization. Prior to LPS stimulation, the application of PSP reversed M1 polarization and noticeably upregulated the expression of the M2 microglial markers, IL-10, CD206, and Arg-1. These results collectively indicated that PSP effectively switched the polarization of LPS-activated BV2 microglia from M1 to a predominantly M2 phenotype. Nonetheless, the precise molecular mechanisms underlying the effect of PSP on microglial polarization remain to be fully elucidated.

The BDNF (brain-derived neurotrophic factor), widely expressed in CNS, is an important regulator of synaptic formation and synaptic plasticity, and also a key point for the brain to regulate emotional behaviors. The neurotrophic hypothesis makes the BDNF an important biomarker of depression. TrkB is a specific receptor for the BDNF, through dimerization and self-phosphorylation, activating intracellular signaling pathways [34]. The ERK pathway (including ERK1 and ERK2) is one of the activated downstream signal pathways. The activation of ERK can induce the phosphorylation of CREB serine 133 residues, produce activated transcriptional complexes, and trigger the activation of target genes [35]. The CREB is a transcription factor that plays an important role in neuronal plasticity and neurogenesis [36]. The CREB and BDNF/TrkB signaling pathways form the positive feedback loop, exerting antidepressant-like effects. In the LPS-induced depression model of rats, the reduced level of the BDNF and the related changes of synaptic plasticity in the hippocampus were observed [37]. Decreased levels of the BDNF and TrkB were found in postmortem brain samples from patients with depression [38]. Ononin treatment significantly decreased depression-like behaviors and activated BDNF/TrkB/CREB signaling pathways in the frontal cortex and hippocampus of chronic mild stress -induced depressive rats [39]. Based on these facts, the BDNF/TrkB/CREB signaling pathway is acknowledged to play a key role in the development and treatment of depression.

Many studies have shown that depression tends to activate the M1 microglia phenotype and inhibit the M2 phenotype. It is speculated that there could be an association between microglial polarization and BDNF/TrkB/CREB signaling in the context of depression. The previous reports revealed that the BDNF was not only expressed in the CNS, but also was released in microglia [40,41]. During the development of the brain, microglia play a vital role in synaptic plasticity by trimming synapses and refining the neural circuit through phagocytosis [42]. The BDNF secreted by microglia is an important material basis for maintaining the formation of the learning-dependent synapse [43]. The BDNF is also an important medium for the information transmission between microglia and neurons. Promoting the M2 phenotype polarization of microglia can directly enhance the synthesis of the BDNF and the expression of neurotrophic receptors, thus protecting neuron survival or preventing neuron apoptosis [44]. In the present study, LPS stimulation remarkably lowered the protein expression level of the BDNF and the phosphorylation levels of TrkB, ERK, and CREB in BV2 cells, while PSP intervention reversed these changes. Thus, M1/M2 polarization regulation and the inhibition of PSP in LPS-induced neuroinflammatory responses might be related to activating the BDNF/TrkB/CREB signaling pathway.

The Notch signaling pathway is mainly composed of receptors, ligands, transcription factors, regulatory molecules, and downstream effector molecules. When the Notch signaling pathway is activated, intracellular domains are released into the cytoplasm and translocated to the nucleus, promoting the production of transcription-activating factors and inducing the expression of downstream target genes such as Hes1 and Hes5 [45,46]. It has been found that the Notch signaling pathway can regulate the polarization of monocyte macrophages. Linc00514 could promote the M2 polarization of tumor-associated macrophages via the Jagged1-mediated Notch signaling pathway [47]. Notch-RBP-J signaling regulates the transcription factor IRF8 to promote inflammatory macrophage polarization [48]. Ganoderma lucidum polysaccharides could repair the chaos in the polarization of M1/M2 macrophages and the molecular mechanism linked to the Notch signaling pathway [49]. Analogous to macrophages, the Notch signaling pathway is also closely related to the activation and polarization of microglia. Studies show that LPS increased the expression of various members of the Notch-1 pathway, including the intracellular Notch receptor domain (NICD), recombining the binding protein suppressor of hairless (RBP-Jκ) and the transcription factor hairy and enhancer of split-1 (Hes-1) in microglia in postnatal rat brains and in BV-2 microglia. The activation of Notch-1 signaling promoted microglia migration and Gastrodin could inhibit the migration of activated BV-2 microglia by regulating the Notch-1 signaling pathway [50]. Simvastatin alters the M1/M2 polarization of murine BV2 microglia via Notch signaling [51]. Lipoxygenase A4 may regulate the microglial polarization after cerebral ischemia-reperfusion injury through the Notch signaling pathway. However, blocking the Notch signaling pathway with the inhibitor DAPT significantly mitigated the effect of LXA4 on microglia differentiation [52]. The evidence presented above shows that the Notch pathway is an important target for regulating the activation, polarization, and inflammatory response of microglia. In the present study, LPS stimulation significantly reduced the protein expression level of Notch1 and Hes1, while PSP intervention reversed these changes. Thus, M1/M2 polarization regulation and the inhibition of PSP in LPS-induced neuroinflammatory responses might be related to modulating the Notch signaling pathway.

Recently, the exploitation and utilization of medicinal plant resources has increased significantly. PSP as traditional food and medicine homologous resources, which are almost nontoxic and rarely have negative effects, have great potential in medical treatment, health care, and the food industry. From our results and those found in the literature, it can be explained that the antidepressant effects of PSP may be attributed, at least in part, to their microglial polarization regulation via the BDNF/TrkB/CREB and Notch/Hes1 signaling pathways. This could provide a reliable basis for the treatment of depression in the future. However, the present study has several limitations. First, the present study uses an in vitro model only. Nevertheless, studies have shown that PSP could reverse the depressive-like behavior in LPS-induced mice by inhibiting the inflammatory response [14]. Second, the current research primarily focuses on the neighboring molecules of the signaling pathways instead of all of them. Further studies will be required to elucidate the full spectrum mechanisms of the microglial polarization regulation of PSP, especially in depression animal models. The relationship between the regulation of microglia polarization by PSP and other antidepressant mechanisms, such as oxidative stress, the HPA axis, monoamine neurotransmitters, tryptophan metabolism, and the microbiota–gut–brain axis, is also worth being included in future investigations, which may ultimately give more ideas about the application of PSP for the treatment of depression in the clinic.

## 5. Conclusions

The current study indicated that PSP facilitates microglial polarization from the pro-inflammatory phenotype towards the anti-inflammatory phenotype, which are associated with the regulation of the BDNF/TrkB/CREB and Notch/Hes1 signaling pathways. This study provides novel insights into fully elucidating the molecular mechanisms of PSP treatment in depression.

## Figures and Tables

**Figure 1 nutrients-16-00438-f001:**
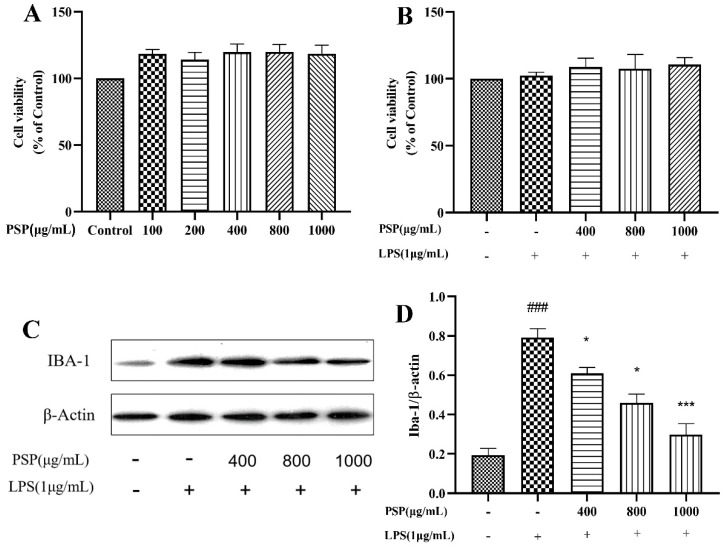
Effects of PSP on the viability and activation of BV2 cells. (**A**) Effects of different concentrations of PSP on cell viability (*p* < 0.001). (**B**) Effects of PSP on cell viability with LPS stimulation (*p* = 0.069). (**C**) Protein expression of Iba-1, representative protein bands. (**D**) The ratio of Iba-1/β-actin (*p* < 0.001). The data are expressed as the means ± SEM. ### *p* < 0.001 versus the control group; * *p* < 0.05, *** *p* < 0.001, versus the LPS-treated group.

**Figure 2 nutrients-16-00438-f002:**
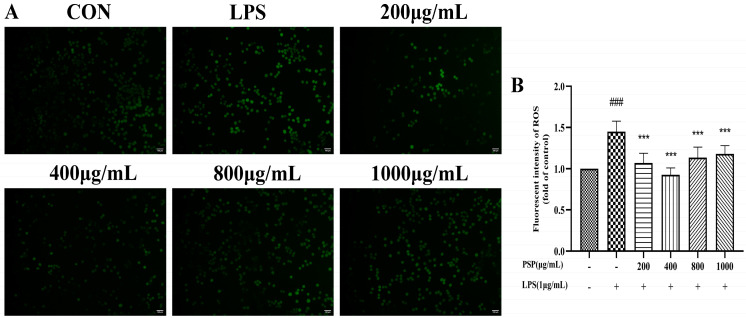
Effects of PSP on ROS release in LPS-induced BV2 cells. (**A**) Representative microscopic images. (**B**) Fluorescence intensity of ROS analyzed with Image J (*p* < 0.001). The data are expressed as the means ± SEM. ### *p* < 0.001 versus the control group; *** *p* < 0.001, versus the LPS-treated group. Scale bar = 100 μm.

**Figure 3 nutrients-16-00438-f003:**
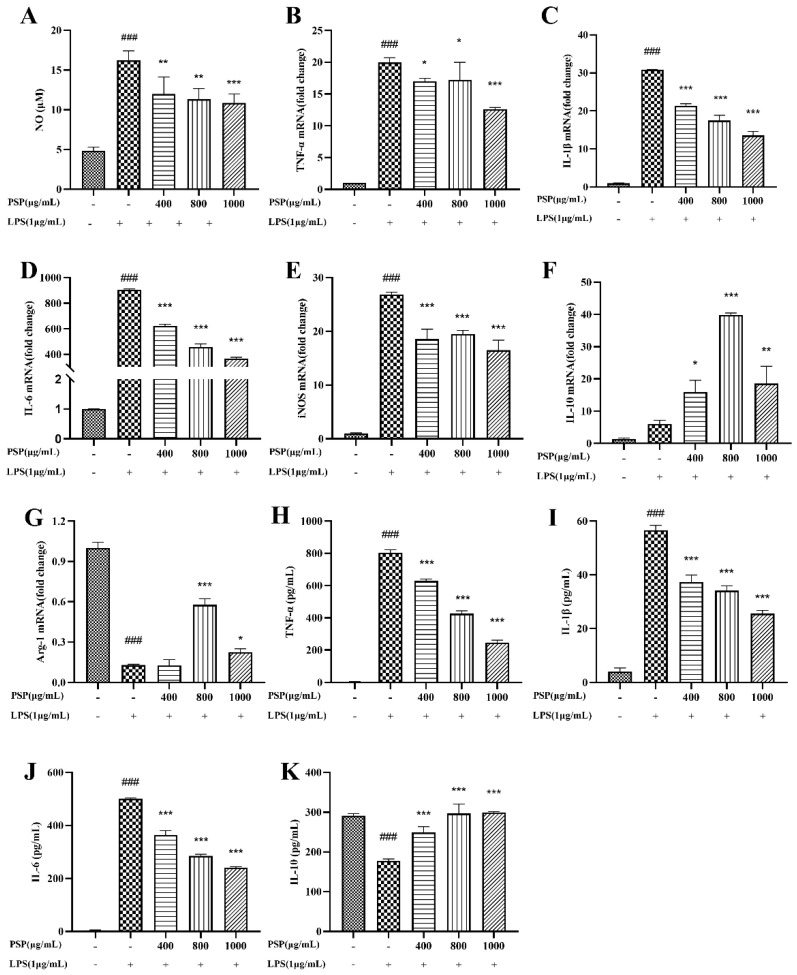
Effects of PSP on LPS-induced NO production and inflammatory cytokine production in BV2 cells. (**A**) Effects of PSP on NO release in LPS-induced BV2 cells (*p* < 0.001). (**B**–**E**) The mRNA expression levels of M1 phenotype genes (TNF-α, IL-1β, IL-6, iNOS) (*p* < 0.001). (**F**,**G**) The mRNA expression levels of M2 phenotype genes (IL-10, Arg-1) (*p* < 0.001). (**H**–**K**) TNF-α, IL-1β, IL-6, IL-10 levels of cell supernatant detected by ELISA (*p* < 0.001). The data are expressed as the means ± SEM. ### *p* < 0.001 versus the control group; * *p* < 0.05, ** *p* < 0.01, *** *p* < 0.001, versus the LPS-treated group.

**Figure 4 nutrients-16-00438-f004:**
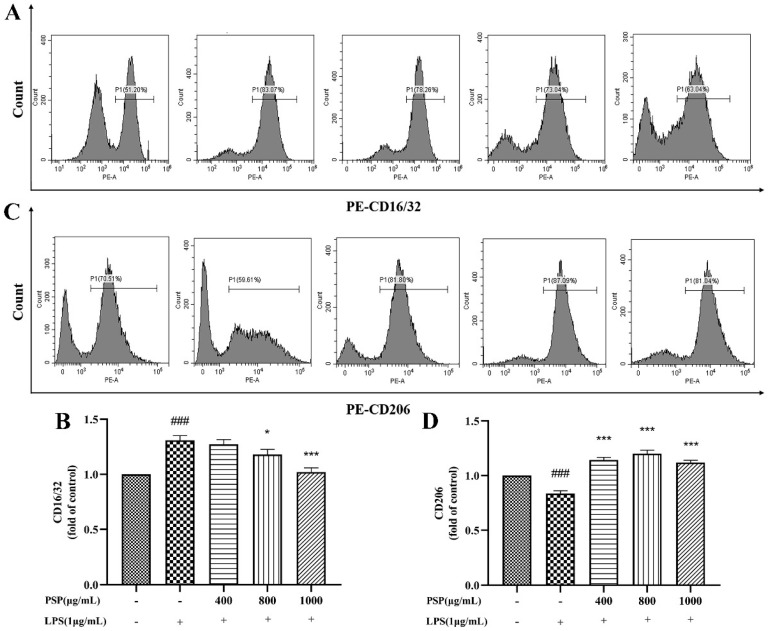
Effects of PSP on LPS-induced protein expression of CD16/32 and CD206 in BV2 cells. (**A**,**B**) CD16/32 (M1) (*p* < 0.001) and (**C**,**D**) CD206 (M2) (*p* < 0.001) protein expression measured by flow cytometry. The data are expressed as the means ± SEM. ### *p* < 0.001 versus the control group; * *p* < 0.05, *** *p* < 0.001, versus the LPS-treated group.

**Figure 5 nutrients-16-00438-f005:**
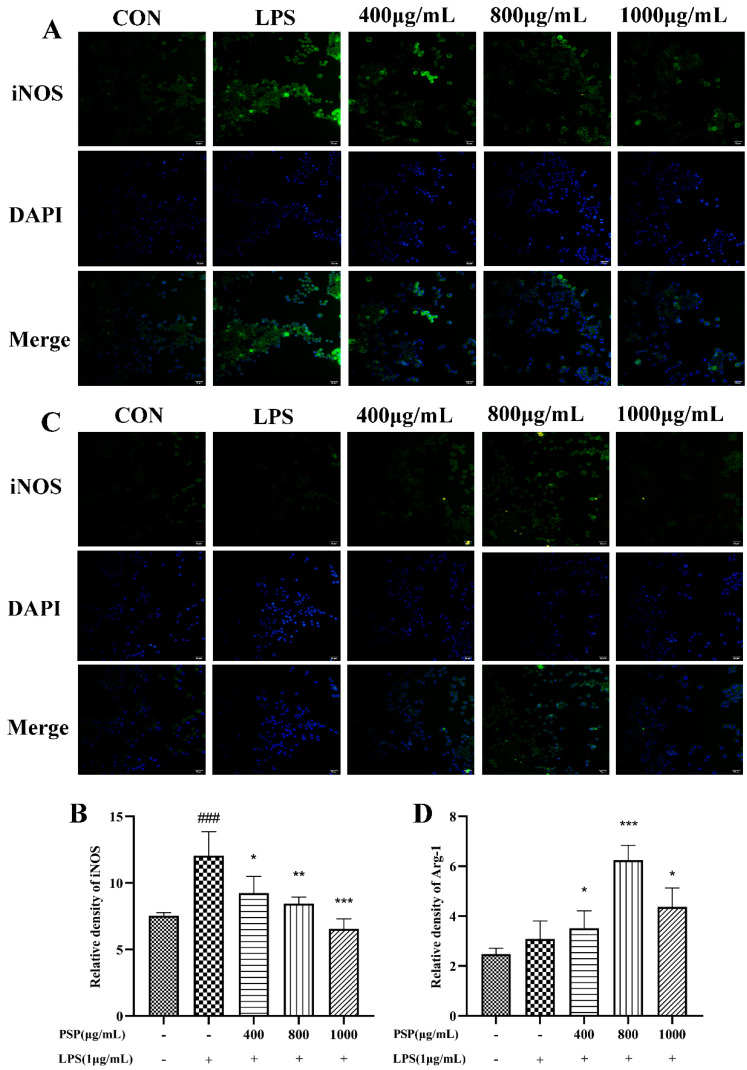
The relative fluorescence intensity of iNOS and Arg-1 in the different groups of BV-2 cells. Representative immunofluorescence picture of (**A**) iNOS (M1) and (**C**) Arg-1 (M2), fluorescence intensity of (**B**) iNOS (M1) (*p* = 0.038) and (**D**) Arg-1 (M2) (*p* = 0.015) analyzed with Image J. The data are expressed as the means ± SEM. ### *p* < 0.001 versus the control group; * *p* < 0.05, ** *p* < 0.01, *** *p* < 0.001, versus the LPS-treated group. Scale bar = 50 μm.

**Figure 6 nutrients-16-00438-f006:**
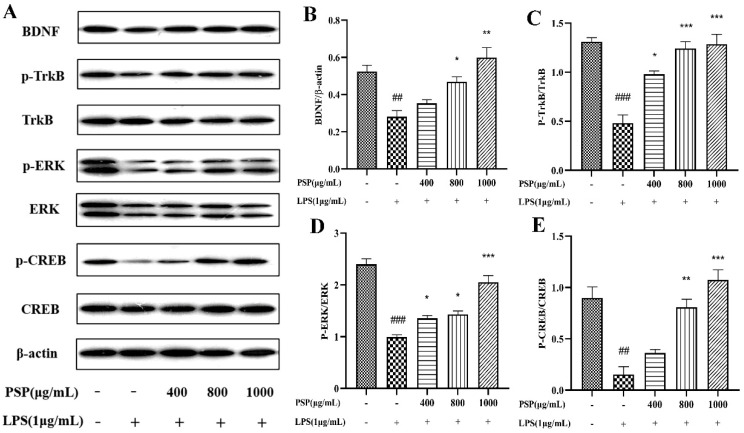
Effect of PSP on BDNF/TrkB/CREB signaling pathway in LPS-induced BV2 cell. (**A**) Representative protein bands, protein expression of (**B**) BDNF (*p* < 0.001). Phosphorylation of (**C**) TrkB (*p* < 0.001), (**D**) ERK (*p* < 0.001), (**E**) CREB (*p* < 0.001). The data are expressed as the means ± SEM. ## *p* < 0.01, ### *p* < 0.001 versus the control group; * *p* < 0.05, ** *p* < 0.01, *** *p* < 0.001, versus the LPS-treated group.

**Figure 7 nutrients-16-00438-f007:**
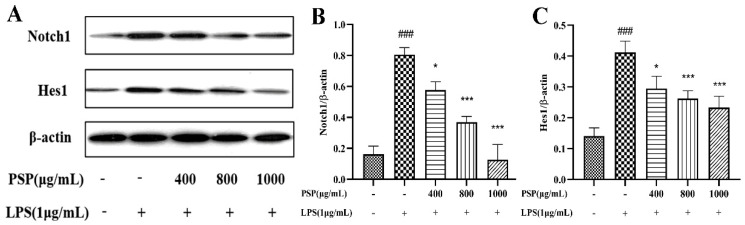
Effect of PSP on Notch signaling pathway in LPS-induced BV2 cell. (**A**) Representative protein bands. Protein expression of (**B**) Notch1 (*p* < 0.001) and (**C**) Hes1 (*p* < 0.001). The data are expressed as the means ± SEM. ### *p* < 0.001 versus the control group; * *p* < 0.05, *** *p* < 0.001, versus the LPS-treated group.

**Table 1 nutrients-16-00438-t001:** Primers for qPCR.

Gene	Sense	Anti-Sense
TNF-α	CACCACCATCAAGGACTCAA	AGGCAACCTGACCACTCTCC
IL-1β	AAATACCTGTGGCCTTGGGC	CTTGGGATCCACACTCTCCAG
IL-6	CCAGAGATACAAAGAAAT	ACTCCAGAAGACCAGAGGAAAT
IL-10	GTGGAGCAGGTGAAGAGTGA	TCGGAGAGAGGTACAAACGAG
iNOS	GAGGCCCAGGAGGAGAGAGATCCG	TCCATGCAGACAACCTTGGTGTTG
CD206	CTTCGGGCCTTTGGAATAAT	TAGAAGAGCCCTTGGGTTGA
Arg-1	GTGAAGAACCCACGGTCTGT	CTGGTTGTCAGGGGAGTGTT
β-actin	GGCTGTATTCCCCTCCATCG	CCAGTTGGTAACAATGCCATGT

## Data Availability

Data are contained within the article.

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
