# Peer review of "Investigating the Antidepressant Mechanisms of Polygonum sibiricum Polysaccharides via Microglial Polarization"

_nutrients, 2024, doi:10.3390/nu16030438_

Round 1

Reviewer 1 Report

Comments and Suggestions for Authors

Please improve the description of the plant, including specialised metabolites, in the introduction.

Please improve the quality of figure 2.

Please include ANOVA P values in figure captions.

Please check all references in the text and format according to journal guidelines.

Please also check the format of the references in the reference section.

Please correct the plant name in the title.

Please include the description of polysaccharide fraction in materials and methods.

Comments on the Quality of English Language

Minor corrections

Author Response

Thanks very much for your hard work in reviewing our manuscript. Please see the attachment.

Reviewer 2 Report

Comments and Suggestions for Authors

The manuscript "Research on the Antidepressant Mechanisms of Polygonati sibiricum Polysaccharides Based on Microglia Cell Polarization" shows a molecular approach trying to elucidate the action of polysaccharides in and its antidepressant activity based microglia cell polarization.

Despite the positive results shown in cell culture, the polysaccharides are not available in the central nervous system to trigger the actions shown in this work.

The second critical point is the concentration that authors used to obtain the positive results in microglia polarization that were in order of mg/mL and I consider extremely high to talk about new therapeutic approaches to handle depression.

I strongly suggest that authors focus on the gut nervous system and its microbiota to explain the digestibility and effect of polysaccharides. For example, the production of 5-HT (5-hydroxytryptamine) can be affected by these polysaccharides polarization M1/M2 in the gut?

Author Response

Thanks very much for your advice. Please see the attachment.

Reviewer 3 Report

Comments and Suggestions for Authors

The paper presents a comprehensive exploration of the role of neuroinflammation and microglial polarization in depression, with a focus on the potential therapeutic effects of Polygonati 2 sibiricum Polysaccharides (PSP).

The introduction highlights the global health problem of depression communicating the urgency of finding new methods and targets for its treatment. The mention of the association between depression and systemic inflammation sets the stage for the study's focus on microglia and polarization. However, it could benefit from a brief transition sentence to smoothly connect the general context of depression to the specific focus on microglia. Moreover, the introduction has a logical flow, presenting the problem, rationale, and context before introducing the study's focus on microglia and PSP.

I propose changing the title to: “Investigating the Antidepressant Mechanisms of Polygonati 2 sibiricum Polysaccharides through (or via) Microglial Cell Polarization”

Addressing the following suggestions will enhance clarity and completeness:

-replace “depression drugs” with “drugs used to treat depression”

-clarify why the BV2 cell line and LPS-induced activation model were selected, and provide more context for their relevance in studying microglial polarization

-provide more information on the rationale behind the concentrations chosen for PSP (400, 800, and 1000 μg/mL)

-specify the units for NO concentration

-explain the 6 hours incubation without LPS in the ROS assay

-mention the scale or range used for the fluorescence microscopy analysis

-specify the units for the measured cytokines in the ELISA

-clarify the concentration of RIPA buffer used

-To enhance the paper further, consider addressing potential limitations, such as the use of an in vitro model, and discussing the relevance of the findings to depression in vivo

-Additionally, emphasizing the clinical implications of PSP in treating depression and suggesting future research directions could enhance the paper's overall impact

-clearly articulate the proposed relationship between PSP, microglial polarization, and the BDNF/TrkB/CREB signaling pathway and discuss existing literature supporting the connection between microglial polarization and BDNF/TrkB/CREB signaling in the context of depression

-similar to the BDNF/TrkB/CREB pathway, provide more context on the relationship between microglial polarization and the Notch/Hes1 signaling pathway. Highlight any existing studies or evidence supporting the involvement of the Notch pathway in microglial polarization

Comments on the Quality of English Language

minor editing

Author Response

(The authors gave the same response as above.)

Round 2

Reviewer 2 Report

Comments and Suggestions for Authors

All issues were addressed well.